# SRAMA: Learning Syllable Representations with Adaptive local Monotonic Attention for Syllable Stress Detection in L2 Speech

## Abstract

Syllable stress plays a key role in stress-timed languages such as English, where shifting prominence/stress (e.g., obJECT vs. OBject) alters meaning and can hinder communication. For second-language (L2) learners, correct placement of syllable stress is a challenge, creating a strong need for automatic stress detection methods in computer-assisted language learning (CALL) systems. Existing methods, however, depend on syllable boundaries, which are costly to manually annotate and error-prone when generated automatically. To overcome this limitation, we propose a boundary-independent framework that learns syllable representations directly from speech and predicts their stress without explicit boundaries. The framework has three main components: (1) an Adaptive local Monotonic Attention (AMA) mechanism that captures syllable representations by enforcing the natural left-to-right order of syllables, (2) contrastive loss that drives stressed and unstressed syllables apart in the embedding space, and (3) an end-to-end encoder–decoder pipeline that integrates these two components to map speech frames to syllable representations and then decode them autoregressively into stress predictions. To demonstrate the effectiveness of each component, we further conduct ablation studies by either removing AMA or contrastive loss, which validate their critical role in learning robust syllable representations. Experiments on the ISLE corpus of German and Italian learners of English show that our method outperforms boundary-dependent baselines, thereby overcoming the need for explicit boundaries in stress prediction.

## 1 Introduction

In stress-timed languages such as English, one syllable in a word is typically pronounced with greater prominence, known as syllable stress Goedemans et al. (2018). This stress is conveyed through acoustic cues such as pitch, duration, and intensity Cutler & Isard (1980) . Shifting the syllable stress in a word can change its meaning, for example, obJECT (verb, meaning "to carry out or perform") versus OBject (noun, meaning "a thing or item"). Correctly perceiving and producing this stress is particularly challenging for second-language (L2) learners, who often transfer stress patterns from their native language. To address such challenges, Computer-Assisted Language Learning (CALL) systems Ferrer et al. (2015) have been developed, providing automated feedback to guide learners toward correct stress placement and improve spoken communication. A core component of these systems is automatic syllable stress detection, which identifies the stress patterns and provide feedback on learners' stress placement.

Over the years, various approaches have been proposed to tackle automatic syllable stress detection which can be broadly categorized into three directions. First, early methods relied on handcrafted acoustic features such as pitch, duration, and intensity Ferrer et al. (2015); Delmonte et al. (1997); Tepperman & Narayanan (2005). These features were often combined with statistical models like decision trees or support vector machines. Second, deep learning models including convolutional and recurrent networks have been used to capture local and sequential patterns from acoustic representations Shahin et al. (2016); Mallela et al. (2023). Third, few works have proposed different loss functions that enforce linguistic constraints, such as ensuring only one primary stressed syllable per word, to further enhance model performance Aluru et al. (2024) ans Aluru et al. (2025) . Despite

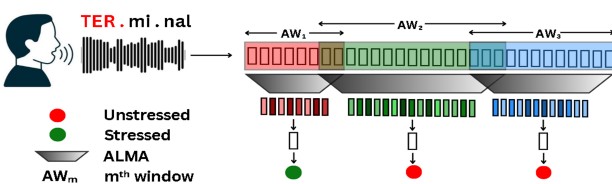

Figure 1: AMA based stress prediction

these advances, a critical bottleneck remains, all of the existing methods rely on syllable or phoneme boundaries. Manual annotation of syllable or phoneme boundaries is costly, and automatic alignment often introduces errors. Consequently, existing boundary-dependent methods can be unreliable when boundaries are unavailable or inaccurate.

To address these limitations, we propose a boundary-independent framework for automatic syllable stress detection. Our work makes three key contributions:

**(1) Adaptive local Monotonic Attention for Learning Syllable Representations:** Syllable stress depends on the sequential structure of syllables within a word, and attending to the entire speech signal at once can mix information across syllables. To address this, we introduce an Adaptive local Monotonic Attention (AMA) mechanism (Figure 1) that progresses through the speech in order, focusing on the segment relevant to the current syllable at each step. Unlike fixed-window attention, our attention window is learnable and adapts for each syllable, allowing the model to dynamically determine how much context is needed for each syllable representation. This ensures that each syllable embedding captures only the most informative segments while preserving the natural left-to-right sequential structure without requiring explicit boundary annotations.

**(2) Contrastive Loss for Discriminative Syllable Representations:** Stressed and unstressed syllables can have overlapping acoustic cues. To improve discriminability, we apply contrastive loss at multiple stages in the network. This encourages stressed and unstressed syllables to occupy distinct regions in the embedding space, helping the model learn robust and discriminative syllable representations that support accurate stress prediction without requiring explicit boundary supervision.

**(3) End-to-End Autoregressive Encoder–Decoder Framework:** We integrate the components into an end-to-end encoder–decoder framework. The encoder generates frame-level representations independently, while the decoder operates autoregressively, predicting stress for each syllable sequentially and using previous syllable embeddings as context. This autoregressive design allows the model to capture sequential dependencies among syllables.

By jointly learning syllable structure and stress patterns in a fully end-to-end manner, the model achieves accurate stress predictions without relying on manually annotated boundaries. We evaluate our method on the ISLE corpus, which consists of non-native (L2) English speech from German and Italian L2 learners. To assess the impact of our approach, we also perform ablation studies comparing models trained with vs without AMA and with vs without contrastive loss, demonstrating that both components substantially improve stress prediction performance.

## 2 RELATED WORK

**Handcrafted Acoustic Feature Approaches:** Early approaches to syllable stress detection relied on handcrafted acoustic features such as intensity, pitch, and duration, often combined with simple statistical models. For instance, Tepperman & Narayanan (2005) used prosodic features derived from vowel duration, intensity contours, and pitch trajectories to evaluate stress patterns, while Yarra et al. (2017) incorporated sonority-based prominence features to better capture the contextual influence of surrounding syllables. Machine learning models such as decision trees and support vector machines (SVMs) were then applied by Johnson & Kang (2015) for syllable stress prediction.

**Deep Learning Approaches:** To move beyond handcrafted features, several deep learning approaches have been explored. Shahin et al. (2016) used convolutional neural networks (CNNs) to extract stress-related patterns from prosodic and spectral features, demonstrating improvements over traditional statistical models. Ruan et al. (2019) employed bidirectional LSTMs using MFCCs, pitch, and energy features to model sequential dependencies across syllables, highlighting the role

of context in stress perception. Mallela et al. (2023) proposed a hybrid approach combining variational autoencoders (VAEs) with deep neural networks (DNNs) to jointly learn rich syllable-level representations, capturing intricate interdependencies among acoustic and contextual features.

**Incorporating Linguistic Constraints:** Many works recognized the importance of linguistic constraints, particularly that only one primary stress occurs per word. In earlier methods, this constraint was often applied as post-processing on the classifier outputs. To directly incorporate such constraints, Aluru et al. (2024) and Aluru et al. (2025) proposed Post-Net and Post-Net2.0 architectures, using time-delay neural networks to model sequential dependencies among syllables.

**Partially Boundary-Independent Methods:** Some recent approaches have sought to reduce reliance on explicit syllable boundaries by using alternative cues such as phoneme boundaries, or latent segment representations for stress prediction. Korzekwa et al. (2020) used dual attention mechanisms, where one attention operates over frame-level features and another over phoneme-level features, allowing the model to focus on informative regions without requiring syllable annotations. While this method mitigate the need for manually annotated syllable boundaries, it still relies on phoneme boundaries.

Although prior works Cho et al. (2024a;b) have explored syllable-level representations but they are not optimized for stress prediction and may miss stressed–unstressed distinctions. In contrast, our method jointly learns syllable representations and its stress in an end-to-end manner, producing embeddings directly informative for stress patterns without boundary annotations.

## 3 PROPOSED METHODOLOGY

### 3.1 FEATURE EXTRACTION AND PREPROCESSING

Self-supervised learning (SSL) features have proven effective in capturing rich acoustic and prosodic information, often outperforming handcrafted features for the tasks like syllable stress detection. In particular, prior work by Mallela et al. (2024) has shown that wav2vec 2.0 encodes cues that are highly relevant for distinguishing stressed from unstressed syllables. Motivated by this, we adopt wav2vec 2.0 for feature extraction. As shown in Figure 2a, for a given utterance the feature extraction block generates 768-dimensional embeddings for each frame, producing a sequence of $t$ frames $\{f_1, f_2, \ldots, f_t\}$.

**Padding:** Since the sequence length ($t$) varies across utterances, this poses a significant challenge for modelling. To address this, we pad all sequences to a fixed length. As shown in Figure 2a, the padding block pads the shorter sequences with zero padding up to $r$ frames, resulting in sequences $\{f_1, f_2, \ldots, f_r\}$. Here, $r$ corresponds to the maximum number of frames in the dataset.

**Masking:** Padding sequences to a fixed length $r$ introduces frames that do not correspond to actual speech, which could negatively impact models' learning. As illustrated in Figure 2a, we use a masking block to mask the zero-padded $r - t$ frames of each utterance.

### 3.2 MODELLING

We describe the main modelling in three components: 1) Encoder to capture contextual information, 2) Adaptive local Monotonic Attention (AMA) for obtaining syllable representations, 3) Autoregressive decoder for syllable stress prediction.

**Encoder:** As shown in Figure 2b, to capture temporal context and dependencies across speech frames, we pass the masked frame-level features through sequential layers in the encoder. This results in a sequence of contextualized frame-level features $\{e_1, e_2, \ldots, e_r\}$. These embeddings provide rich contextual information, which can be selectively aggregated for each syllable by the attention mechanism.

**Adaptive local Monotonic Attention (AMA):** Syllable stress is inherently sequential, and attending over the entire speech signal can mix information across syllables. Inspired by the window attention mechanism for speech recognition Zhang et al. (2019), we introduce an Adaptive local Monotonic Attention (AMA) mechanism for syllable stress detection.

Figure 2: Block diagram of the proposed boundary-independent AMA based syllable stress detection

As illustrated in Figure 2b, only the first $t$ frames of the encoder outputs $(e_1, e_2, \ldots, e_t)$ are considered, as the remaining $r - t$ frames are zero-padded. These $t$ contextualized representations are passed through a windowing block to generate $n$ *local* windows, one per syllable. Unlike fixed-window attention, both the center and spread of each window are *adaptable*, allowing the model to determine the optimal context for each syllable *monotonic*ally. Specifically, for the $m$-th syllable $(S_m)$, the center $(\mu_m)$, adaptive window size $(AW_m)$, and spread of the window $(\sigma_m)$ are defined as:

$$\mu_m = \frac{m + 0.5}{n} \cdot t, \qquad AW_m = (\gamma \cdot t) + \beta, \qquad \sigma_m = \frac{AW_m}{2}, \qquad (1)$$

where $t$ is the total frames, $\gamma$ a learnable scale, $\beta$ a learnable offset. Frames outside each adaptive local window are masked, and attention is computed only over the frames within this window to produce syllable-level representations. Within each adaptive local window, we employ Bahdanau-style additive *attention* Bahdanau et al. (2014), calculating the score between the decoder hidden state and encoder output, and then deriving the attention weights and context vector as:

$$\text{score}_i = v^\top \tanh(W_e e_i + W_h h_m + b), \quad \alpha_i = \frac{\exp(\text{score}_i)}{\sum_{j \in \mathcal{AW}_m} \exp(\text{score}_j)}, \quad S_m = \sum_{i \in \mathcal{AW}_m} \alpha_i e_i. \qquad (2)$$

where, $score_i$ denotes the score for $i^{th}$ encoder output $(e_i)$, $h_m$ is the decoder hidden state, $\alpha_i$ is the attention weights, $S_m$ is the context vector ($m^{th}$ syllable) and $W_e$, $W_h$, $b$, and $v$ are learnable parameters.

**Autoregressive Decoder:** Stress is not independent across syllables; the stress of a syllable often depends on its neighbors. To capture these inter-syllable dependencies, we employ an autoregressive decoder, where the prediction for the current syllable conditions on both its embedding and the predictions of previous syllables. As illustrated in Figure 2b, syllable representations produced by AMA are fed into the decoder, which generates stress labels one at a time. During training, we use teacher forcing to provide the true stress labels of previous syllables, guiding the decoder, while at inference, it relies on its own past predictions. To further enhance the discriminability of stressed versus unstressed syllables after autoregressive processing, we apply a second contrastive loss at the final decoder layer. This ensures that the final stress predictions remain distinct across syllables.

### 3.3 LOSS FUNCTIONS

To make the syllable-level representations from AMA effective, they must be well separated in the embedding space. So, we employ contrastive loss. Further, stress detection is a binary classification task, we also consider binary cross-entropy (BCE) loss for accurate stress label prediction.

**Contrastive Loss:** Contrastive loss encourages similar embeddings to be closer and dissimilar embeddings to be farther. For a pair of representations $(S_i, S_j)$ and a label $y \in \{0, 1\}$ indicating whether both belong to the same stress class, contrastive loss is defined as:

$$L_{\text{contrastive}} = y \cdot \|S_i - S_j\|^2 + (1 - y) \cdot \max(0, m - \|S_i - S_j\|)^2, \tag{3}$$

where $m$ is a margin hyperparameter. In our model, we employ two contrastive losses at different stages: one, applied directly to the syllable embeddings obtained from AMA, and the other applied at the decoder's final layer after autoregressive processing. The first loss ensures that intermediate syllable representations are well separated, while the second reinforces this separation in the final representations used for stress prediction. Together, these two losses provide multi-stage supervision, guiding the model to produce distinct, stress-aware embeddings throughout the network.

**Binary Cross-Entropy Loss:** The primary objective is to correctly classify each syllable as stressed or unstressed. For a predicted label $\hat{y}$ and ground truth label $y \in \{0, 1\}$, the binary cross-entropy (BCE) loss is defined as:

$$L_{\text{BCE}} = -\big[y \log(\hat{y}) + (1 - y) \log(1 - \hat{y})\big]. \tag{4}$$

This ensures that the decoder assigns high probability to the correct stress label for each syllable.

**Total Loss:** The final training loss is:

$$L_{\text{total}} = L_{\text{BCE}} + \lambda_1 L_{\text{contrastive}}^{\text{attention}} + \lambda_2 L_{\text{contrastive}}^{\text{decoder}}, \tag{5}$$

where $\lambda_1$ and $\lambda_2$ balances the contribution of the contrastive losses relative to the classification objective.

### 3.4 POST-PROCESSING

In English, each word typically contains a single primary stressed syllable, known as linguistic constraint. However, model predictions can sometimes assign stress to multiple syllables or to none. As shown in Figure 2c, we apply a post-processing step (non trainable) to enforce the linguistic constraint. For each word, the syllable with the highest predicted stress probability from the decoder is selected as stressed, while all other syllables are set as unstressed.

## 4 DATASET

For our experiments, we use the ISLE corpus Menzel et al. (2000), which contains 7,834 utterances from 46 non-native English speakers, evenly split between German (GER) and Italian (ITA) learners. Each speaker contributed roughly 160 utterances. Initial phoneme-level alignments were generated using automatic forced alignment and then manually refined by a team of five expert linguists. Syllabification was performed with the P2TK toolkit Tauberer (2008), followed by manual annotation of syllable-level stress according to the English phonological constraint that each word contains exactly one primary stressed syllable. For our study, we only utilize the audio recordings and the syllable-level stress labels, without relying on phoneme or syllable boundaries. The full dataset contains 48,868 stressed and 16,693 unstressed syllables. By following Yarra et al. (2017), we focused only on polysyllabic words (words with two or more syllables) which filtered the dataset to 12,388 stressed and 16,005 unstressed syllables. Speaker-independent train/test splits are created separately for the GER and ITA subsets, balanced across speaker attributes such as age, sex, and proficiency, while preserving the natural distribution of stressed and unstressed syllables.

## 5 EXPERIMENTAL SETUP AND ABLATION STUDIES

We first describe the architectural details of our proposed model, followed by an overview of the baselines used for comparison. Finally, we present the ablation studies conducted to demonstrate the contributions of individual components in our approach.

## 5.1 ARCHITECTURE DETAILS

**Encoder:** A bidirectional LSTM (Schuster & Paliwal (1997)) followed by three ReLU-activated linear layers maps 768-dimensional Wav2Vec 2.0 features to 16-dimensional contextualized representations.

**AMA:** The $\gamma$ parameter of windowing block is set to 0.3 initially.

**Decoder:** The decoder comprises a single GRU layer followed by two linear layers: one for mapping syllable embeddings to stress labels, and other for generating embeddings used in contrastive loss. The number of syllables per utterance, $n$, is known from the data.

**Loss Functions:** We set the $\lambda_1 = 1$ and $\lambda_2 = 1$ to balance both the contrastive losses.

**Training Details:** We train the model using the Adam optimizer with a learning rate of 0.0001 and a batch size of 32, employing early stopping over 100 epochs. Five-fold cross-validation is performed, and experiments are conducted under three scenarios:

**Matched:** In this scenario, the model is trained and tested on data from speakers of the same language (either GER or ITA). This setup evaluates the model's ability to capture syllable-level stress patterns when the training and test distributions are identical.

**Combined:** Here, the model is trained on a mixture of data belongs to both languages (GER + ITA) and tested separately on each language. This setup assesses the model's capacity to generalize across languages when it has seen multilingual data during training.

**Cross:** In this case, the model is trained on data of one language and tested on the other (e.g., trained on GER, tested on ITA, and vice versa). This scenario evaluates the model's ability to generalize stress patterns across languages without exposure to the target language during training.

## 5.2 BASELINES

**Boundary-Dependent Baseline:** As boundary-dependent baselines, we adopt the methods described by Mallela et al. (2024), which employ DNN and LSTM architectures for stress classification. These approaches provide a strong reference point since they rely on accurate syllable boundary information, allowing precise alignment of acoustic cues with the corresponding syllables.

**Partially Boundary-Independent Baseline:** To partially reduce dependence on explicit syllable boundaries, we adopt *attTTS* by Korzekwa et al. (2020) as a partially boundary-independent baseline. This model applies dual attention mechanisms: one over frame-level features and another over phoneme-level embeddings. By comparing our proposed boundary-independent method against attTTS, we evaluate the effectiveness of learning syllable representations without relying on either phoneme or syllable boundaries.

## 5.3 ABLATION STUDIES

We perform the following comprehensive ablation studies to evaluate the impact of each component in our proposed architecture:

**Effect of AMA:** We compare the full model with a variant in which the AMA mechanism is removed. In this variant, the decoder attends over all encoder outputs without focusing on adaptive local windows. This ablation helps us to evaluate the hypothesis that attending to syllable-localized windows produces more discriminative embeddings and improves stress prediction. Without AMA, information from multiple syllables can mix, potentially degrading both embedding quality and stress classification accuracy.

**Effect of Contrastive Loss:** To assess the importance of embedding separation, we train a variant of the model with both contrastive losses removed, i.e., neither on the embeddings from AMA nor on the decoder output embeddings. This ablation evaluates whether enforcing separability at multiple representation levels improves the model's ability to distinguish stressed from unstressed syllables. We believe without contrastive loss, embeddings tend to be less structured, which can increase classification errors.

Table 1: Syllable-level stress detection accuracies with and without (in brackets) post-processing on GER and ITA under different experimental scenarios

| | | Baselines | | | Partially Boundary - independent |
|---|---|---|---|---|---|
| | | Boundary - dependent | | Partially Boundary - independent | |
| | Accuracy | DNN | LSTM | AttTTS | |
| GER | Matched | **94.04** (91.32) | 93.12 (91.70) | NA 70.40 | 93.73 **(93.7)** |
| | Combined | 94.09 (92.93) | 93.77 (92.36) | NA 76.19 | **95.11** **(95.02)** |
| | Cross | 90.74 (87.33) | 89.98 (88.31) | NA 73.13 | **92.77** **(92.7)** |
| ITA | Matched | 93.61 (91.13) | 93.14 (91.68) | NA 77.13 | **95.81** **(95.68)** |
| | Combined | 94.33 (92.26) | 93.79 (92.15) | NA 80.12 | **96.61** **(96.55)** |
| | Cross | 92.09 (88.89) | 91.369 (90.41) | NA 73.10 | **93.8** **(93.77)** |

**Effect of adaptable Window spread ($\sigma$):** To evaluate the benefit of adaptively learning the window size, we conduct an ablation in which the window spread $\sigma$ is non-adaptable instead of being adaptable. This tests whether allowing the model to adjust the window spread for each syllable improves the quality of syllable-level representations and stress predictions. We hypothesize that fixing $\sigma$ can limit the model's flexibility to capture variable-length syllables, potentially reducing classification accuracy.

# 6 RESULTS AND DISCUSSION

In this section, we (i) compare our approach with boundary-dependent and boundary-independent baselines, (ii) assess the effect of the AMA mechanism, (iii) analyze the impact of contrastive loss, and (iv) attention heatmaps analysis for a) AMA vs. no AMA and b) adaptable vs. fixed $\sigma$.

## 6.1 COMPARISON OF PROPOSED APPROACH WITH BASELINES

Table 1 presents syllable-level stress detection accuracies for the proposed boundary-independent approach in comparison with boundary-dependent and partially boundary-independent baselines across matched, combined, and cross conditions for the German (GER) and Italian (ITA) subsets.

For the GER subset, our approach achieves 93.73%, 95.11%, and 92.77% accuracy in the matched, combined, and cross scenarios, respectively. For ITA, it reaches 95.81%, 96.61%, and 93.8%. These results consistently surpass previous methods, highlighting the robustness of our architecture.

**Boundary-dependent baselines:** Compared to DNN and LSTM models that rely on precise syllable boundaries, our approach provides absolute improvements of +1.02% and +2.03% for GER in the combined and cross scenarios, while it is slightly below DNN in the matched setting -0.31%. For ITA, we observe gains of +2.20%, +2.28%, and +1.71%.

**Partially boundary-independent baselines:** Relative to AttTTS, which also does not use explicit syllable boundaries, our method shows marked improvements. For GER, the absolute gains are +23.33%, +18.92%, and +19.64%; for ITA, they are +18.68%, +16.49%, and +20.7% across the three scenarios.

These results show that our boundary-independent approach outperforms prior methods, demonstrating the effectiveness of AMA and contrastive representation separation.

## 6.2 EFFECT OF AMA

To evaluate the contribution of the AMA mechanism, we compare our model with a variant in which AMA is removed. In this analysis, in order to isolate effect of AMA, we do not use contrastive loss

for either of the variant. In the variant without AMA, the decoder attends over all encoder outputs without focusing on syllable-localized windows. As shown in Table 2, adding AMA consistently improves syllable-level stress detection across both GER and ITA subsets:

**GER:** Accuracy increases from 91.51% to 93.37% in the matched scenario (+1.86%), from 92.6% to 94.56% in the combined scenario (+1.96%), and from 89.69% to 90.63% in the cross scenario (+0.94%).

**ITA:** Accuracy rises from 92.88% to 93.96% in matched (+1.08%), from 94.12% to 94.65% in combined (+0.53%), and from 90.19% to 93.16% in cross (+2.97%).

These results show that AMA guides the decoder to syllable-localized features, enhancing syllable-level representations and stress prediction accuracy, with particularly strong gains in cross-lingual scenarios.

Table 2: Accuracies with and without (in bracket) post-processing of models trained with & without AMA

| | Accuracy | | without AMA block | with AMA block |
|---|---|---|---|---|
| | | Matched | 91.51 (91.43) | **93.37** (**93.35**) |
| | | Combined | 92.6 (92.59) | **94.56** (**94.54**) |
| GER | | Cross | 89.69 (89.63) | **90.63** (**90.6**) |
| | | Matched | 92.88 (92.91) | **93.96** (**93.9**) |
| | | Combined | 94.12 (94.09) | **94.65** (**94.65**) |
| ITA | | Cross | 90.19 (90.17) | **93.16** (**93.14**) |

Table 3: Accuracies with and without (in bracket) post-processing of models trained with & without Contrastive Loss

| | Accuracy | | without contrastive loss | with contrastive loss |
|---|---|---|---|---|
| | | Matched | 93.37 (93.35) | **93.73** (**93.7**) |
| | | Combined | 94.56 (94.54) | **95.11** (**95.02**) |
| GER | | Cross | 90.63 (90.6) | **92.77** (**92.7**) |
| | | Matched | 93.96 (93.9) | **95.81** (**95.68**) |
| | | Combined | 94.65 (94.65) | **96.61** (**96.55**) |
| ITA | | Cross | 93.16 (93.14) | **93.8** (**93.77**) |

## 6.3 EFFECT OF CONTRASTIVE LOSS

To investigate the impact of embedding separation on stress prediction, we compare our model trained with and without the contrastive loss, while keeping AMA enabled in both cases because of its effectiveness as discussed in section 6.2. As shown in Table 3, including contrastive loss consistently improves performance across both GER and ITA subsets:

**GER:** Accuracy increases from 93.37% to 93.73% in the matched scenario (+0.36%), from 94.56% to 95.11% in combined (+0.55%), and from 90.63% to 92.77% in cross (+2.14%).

**ITA:** Accuracy rises from 93.96% to 95.81% in matched (+1.85%), from 94.65% to 96.61% in combined (+1.96%), and from 93.16% to 93.8% in cross (+0.64%).

These results show that contrastive loss structures the embedding space, separating stressed and unstressed syllables, and complements AMA to improve syllable-level representation and stress prediction accuracy.

## 6.4 ATTENTION HEATMAP ANALYSIS

This section compares two sets of attention heatmaps from the test data: one showing adaptable vs non-adaptable $\sigma$ for the local window, and the other illustrating the effect of with vs without AMA.

### 6.4.1 ADAPTABLE VS FIXED WINDOW SIGMA ANALYSIS

As seen in Figures 3a and 3b for the word CORDUROY, the attention patterns differ notably between adaptable and fixed $\sigma$. The black boxes in each figure indicate the frames spanned by the attention for each syllable. For the first syllable (*K AO R*, boundary frame 14), non-adaptable $\sigma$ attends roughly the first 20 frames, while adaptable $\sigma$ is more focused, spanning up to frame 16. The second syllable (*D AX*, boundary frame 29) attends frames 9–29 with adaptable $\sigma$, versus 3–34 with fixed $\sigma$. The third syllable (*R IY*, boundary frame 45) attends frames 20–41 in the adaptable

case, compared to 14–43 with fixed $\sigma$. While each syllable slightly overlaps with the previous one to capture transitions, adaptable $\sigma$ restricts attention to relevant frames, whereas fixed $\sigma$ produces broader, less targeted attention.

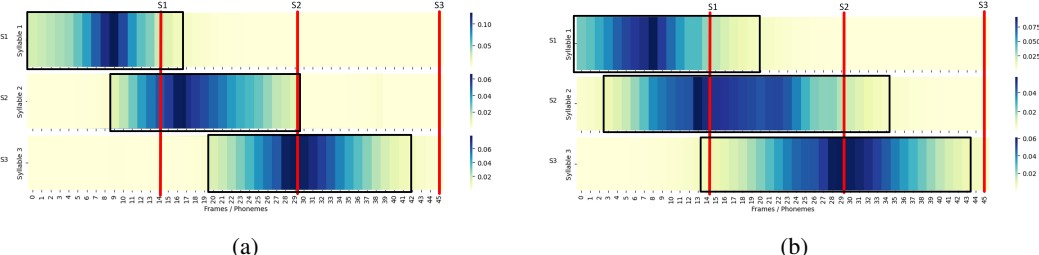

(a)                                                                                    (b)

Figure 3: Attention heatmaps for the word *CORDUROY* with three syllables: *K AO R*, *D AX*, and *R IY*. Red vertical lines denote syllable boundaries. The x-axis shows input frames and the y-axis shows syllable positions. (a) Model with adaptable $\sigma$, (b) Model with non-adaptable $\sigma$.

### 6.4.2   AMA vs No AMA

As shown in Figures 4a and 4b, attention heatmaps for the word *ESTIMATE* highlight clear differences with and without AMA. With AMA, the model focuses on localized syllable-specific regions. Whereas, in without AMA the decoder largely focuses on the early frames of the word for all three syllables, giving slightly different weights to each syllable but not separating them clearly. This highlights the importance of the proposed AMA.

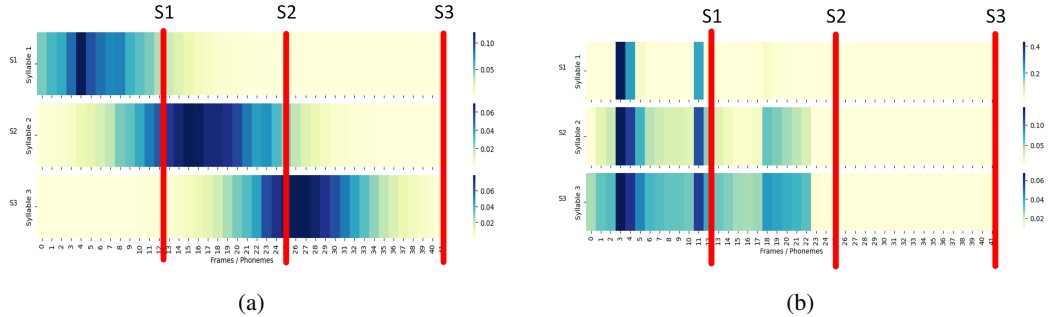

(a)                                                                                    (b)

Figure 4: Attention heatmaps for the three-syllable word *ESTIMATE*, with syllables *EH*, *S T IH*, and *M EY T*. (a) Model trained with AMA, (b) Model trained without AMA. Here, the red vertical lines denote syllable boundarie

## 7   Conclusion

In this study, we proposed a boundary-independent approach for syllable-level stress detection that combines Adaptive local Monotonic Attention (AMA) with contrastive embedding learning. By generating localized syllable representations and enforcing separability in the embedding space, our model effectively captures contextual and sequential dependencies without relying on explicit syllable boundaries. Extensive experiments on the ISLE corpus demonstrate that our method consistently outperforms both boundary-dependent and partially boundary-independent baselines across matched, combined, and cross-lingual scenarios. Ablation studies and attention heatmap analyses further highlight the critical contributions of adaptable window spreads, AMA and contrastive loss.

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
