# OpenReview forum: "SRAMA: LEARNING SYLLABLE REPRESENTATIONS WITH ADAPTIVE LOCAL MONOTONIC ATTENTION FOR SYLLABLE STRESS DETECTION IN L2 SPEECH"
_ICLR.cc/2026/Conference — ICLR 2026 Conference Withdrawn Submission_

### Official Review · Reviewer_Aq5F · 2025-10-29

**Soundness:** 3
**Presentation:** 2
**Contribution:** 3
**Rating:** 4
**Confidence:** 3

**Summary:**

The paper proposes SRAMA, a boundary-independent framework for syllable stress detection in L2 speech, integrating Adaptive local Monotonic Attention (AMA), multi-stage contrastive loss, and an end-to-end autoregressive encoder-decoder. It avoids reliance on explicit syllable/phoneme boundaries, achieves better performance than boundary-dependent and partially boundary-independent baselines on the ISLE corpus (German/Italian L2 English learners), and verifies core components via ablation studies.

**Strengths:**

The work addresses the critical "boundary dependence" bottleneck in existing stress detection methods with innovative AMA (dynamic adaptive windows for syllable-localized attention) and multi-stage contrastive loss (solving acoustic overlap of stressed/unstressed syllables). The experimental design is rigorous, covering matched/combined/cross scenarios and sufficient ablation studies to validate component contributions.

**Weaknesses:**

Only validated on ISLE corpus with German/Italian learners, no testing on other L2 groups or stress-timed languages; baselines is relatively outdated.

**Questions:**

Since the paper emphasizes that syllables have sequential dependencies, why is a unidirectional autoregressive model chosen instead of bidirectional modeling (e.g., BERT-style)? Bidirectional modeling should capture both preceding and subsequent syllable contexts, which may better model inter-syllable dependencies.

---

### Official Review · Reviewer_ZVai · 2025-10-30

**Soundness:** 3
**Presentation:** 2
**Contribution:** 2
**Rating:** 2
**Confidence:** 4

**Summary:**

This paper proposes SRAMA, a boundary-independent framework for syllable stress detection in L2 speech. The key contributions are:
1.It introduces an Adaptive local Monotonic Attention (AMA) mechanism that captures syllable representations by enforcing the natural left-to-right order of syllables without relying on explicit boundaries.
2.It employs a contrastive loss that drives stressed and unstressed syllables apart in the embedding space at multiple stages.
3.It establishes an end-to-end encoder-decoder pipeline that integrates AMA and contrastive learning to map speech frames to syllable representations and then decode them autoregressively into stress predictions.
Extensive experiments on the ISLE corpus show that the method outperforms boundary-dependent baselines, thereby overcoming the need for explicit boundaries in stress prediction. Ablation studies validate the critical role of both the AMA mechanism and the contrastive loss.

**Strengths:**

The motivation of this work is highly justified. In computer-assisted language learning (CALL) systems, syllable stress plays a key role. By jointly learning syllable representations and stress patterns in a fully end-to-end manner, the model achieves accurate stress prediction without relying on manually annotated boundaries. From this perspective, the proposed idea holds significant practical implications.

**Weaknesses:**

1、The logical flow of the article's content is weak. Several later sections are presented in a fragmented, point-by-point manner without a unified overall narrative, making it difficult to grasp the complete picture during reading.
2、The Results and Discussion section is filled with a mere enumeration of experimental data, failing to provide clear explanations for the underlying reasons behind the experimental outcomes.
3、The methodology employed in the article is relatively conventional, and the absence of open-source code makes it impossible to verify the effectiveness of the proposed framework.

**Questions:**

1、Restructure the article's organization, ensuring each section begins with a cohesive overview.
2、In the discussion of experimental results, avoid merely listing findings; instead, provide a detailed analysis of the reasons behind particularly notable outcomes, whether positive or negative.

---

### Official Review · Reviewer_d17P · 2025-10-31

**Soundness:** 3
**Presentation:** 3
**Contribution:** 2
**Rating:** 2
**Confidence:** 4

**Summary:**

This paper focuses on the task of syllable stress classification (stressed vs. non-stressed). The authors proposed an Adaptive local Monotonic Attention (AMA) technique which does not require syllable boundaries to make predictions. Experimental results show that even without syllable boundaries, the proposed model can outperform the baseline that uses syllable boundaries.

**Strengths:**

Presentation is clear. The analyses of the method are reasonable.

**Weaknesses:**

1. Although not requiring syllable boundaries, from my understanding, the proposed method still requires knowing the total number of syllables in the input utterance beforehand, as indicated by $m$ and $n$ in equation 1. This will limit the usage in practice.
2. In addition, the window length is controlled only be scalers $\beta$ and $\gamma$, so essentially the window will be static across utterances. I suspect this formulation will negatively impact the generalization to different speakers, speaking rates, rhythm, etc, since in reality there are huge variation on speaking styles.
3. The training and testing data is limited in size and coverage. It is not clear to me the proposed static windowing approach can work well on data in the wild.
4. Overall, I think the technical innovation of this work is limited. This work's focus in narrow on a specific syllable stress prediction task, and the proposed technique AMA already exists. The contribution from algorithmic advancement does not meet the standard of a venue like ICLR.

**Questions:**

See the Weakness session above.

---

### Note · Authors · 2025-11-22

I have read and agree with the venue's withdrawal policy on behalf of myself and my co-authors.